# A State-Vector Framework for Dataset Effects

**Esmat Sahak**[1,2], **Zining Zhu**[1,2], **Frank Rudzicz**[1,2,3]

[1] University of Toronto, [2] Vector Institute for Artificial Intelligence, [3] Dalhousie University

esmat.sahak@mail.utoronto.ca, zining@cs.toronto.edu, frank@dal.ca

## Abstract

The impressive success of recent deep neural network (DNN)-based systems is significantly influenced by the high-quality datasets used in training. However, the effects of the datasets, especially how they interact with each other, remain underexplored. We propose a state-vector framework to enable rigorous studies in this direction. This framework uses idealized probing test results as the bases of a vector space. This framework allows us to quantify the effects of both standalone and interacting datasets. We show that the significant effects of some commonly-used language understanding datasets are characteristic and are concentrated on a few linguistic dimensions. Additionally, we observe some "spill-over" effects: the datasets could impact the models along dimensions that may seem unrelated to the intended tasks. Our state-vector framework paves the way for a systematic understanding of the dataset effects, a crucial component in responsible and robust model development.

## 1 Introduction

In recent years, data-driven systems have shown impressive performance on a wide variety of tasks and massive, high-quality data is a crucial component for their success (Brown et al., 2020; Hoffmann et al., 2022). Currently, the availability of language data grows much more slowly than the computation power (approximately at Moore's law), raising the concern of "data exhaustion" in the near future (Villalobos et al., 2022). This impending limitation calls for more attention to study the quality and the effects of data.

The data-driven systems gain linguistic abilities on multiple levels ranging from syntax, semantics, and even some discourse-related abilities during the training procedures (Liu et al., 2021). The training procedures almost always include multiple datasets – usually there is a "pre-training" phase and a "fine-tuning" phase, where the model devel-

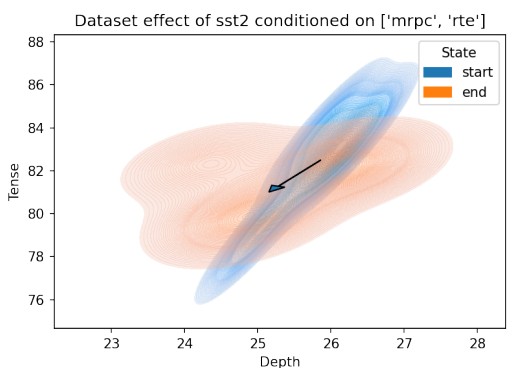

Figure 1: An example of the *individual* dataset effect. This figure shows how two linguistic abilities of RoBERTa, as characterized by the accuracies of two probing tasks: syntactic tree depth and tense, are changed by fine-tuning the SST2 dataset.

opers apply different datasets. Model developers use large corpora that may include multiple existing datasets (Sun et al., 2021; Wei et al., 2021; Brown et al., 2020). How these data relate to the progress in linguistic ability is not systematically studied yet. Each dataset has desired effects, but does it have some under-specified effects? When using multiple datasets together, do they have undesired, interaction effects? These questions become more contingent as the engineering of datasets becomes more essential, yet no existing framework allows convenient quantification of these dataset effects.

Probing provides convenient frameworks to study the linguistic abilities of DNN systems from multiple perspectives. Probing analyses show how the components of DNNs demonstrate linguistic abilities (Tenney et al., 2019; Rogers et al., 2020; Belinkov, 2021; Niu et al., 2022). The probing results are relevant to the model's ability to the extent that the probing accuracies can predict the model's downstream performance (Zhu et al., 2022a). These findings make probing classification a promising candidate for setting up a framework to describe the effects of datasets.

In this paper, we set up a state-vector framework for describing the dataset effects. We formalize idealized probes, which give the true linguistic ability of the DNN model in a state of training. Then, we derive two terms, individual and interaction effects, that describe the dataset effects along multiple dimensions of linguistic abilities. A benefit of our state-vector framework is that it allows a convenient setup of statistical tests, supporting a rigorous interpretation of how the datasets affect the models.

The state framework allows us to frame transitions and set up experiments efficiently. Many frequently-used datasets have "spill-over" effects besides the purposes they are originally collected to achieve. Additionally, the interaction effects are concentrated and characteristic. Our framework provides a systematic approach to studying these effects of the datasets, shedding light on an aspect of model developments that deserves more attentions. All scripts, data, and analyses are available at our GitHub repository.

## 2   Related Works

**Understanding the datasets**   Recent work has investigated various properties of datasets. Swayamdipta et al. (2020) used the signals of the training dynamics to map individual data samples onto "easy to learn", "hard to learn", and "ambiguous" regions. Ethayarajh et al. (2022) used an aggregate score, predictive $\mathcal{V}$-information (Xu et al., 2020), to describe the difficulty of datasets. Some datasets can train models with complex decision boundaries. From this perspective, the complexity can be described by the extent to which the data samples are clustered by the labels, which can be quantified by the Calinski-Habasz index. Recently, Jeon et al. (2022) generalized this score to multiple datasets. We also consider datasets in an aggregate manner but with a unique perspective. With the help of probing analyses, we are able to evaluate the effects of the datasets along multiple dimensions, rather than as a single difficulty score.

**Multitask fine-tuning**   Mosbach et al. (2020) studied how fine-tuning affects the linguistic knowledge encoded in the representations. Our results echo their finding that fine-tuning can either enhance or remove some knowledge, as measured by probing accuracies. Our proposed dataset effect framework formalizes the study in this direction. Aroca-Ouellette and Rudzicz (2020) studied the effects of different losses on the DNN models and used downstream performance to evaluate the multidimensional effects. We operate from a dataset perspective and use the probing performance to evaluate the multidimensional effects. Weller et al. (2022) compared two multitask fine-tuning settings, sequential and joint training. The former can reach higher transfer performance when the target dataset is larger in size. We use the same dataset size on all tasks, so either multitask setting is acceptable.

## 3   A State-Vector framework

This section describes the procedure for formulating a state-vector framework for analyzing dataset effects in multitask fine-tuning.

**An abstract view of probes**   There have been many papers on the theme of probing deep neural networks. The term "probing" contains multiple senses. A narrow sense refers specifically to applying *post-hoc* predictions to the intermediate representations of DNNs. A broader sense refers to examination methods that aim at understanding the intrinsics of the DNNs. We consider an abstract view, treating a probing test as a map from multidimensional representations to a scalar-valued test result describing a designated aspect of the status of the DNN. Ideally, the test result faithfully and reliably reflects the designated aspect. We refer to the probes as *idealized* probes henceforth.

**Idealized probes vectorize model states**   Training DNN models is a complex process consisting of a sequence of states. In each state $S$, we can apply a battery of $K$ *idealized* probes to the DNN and obtain a collection of results describing the linguistic capabilities of the model's state at timestep $\mathcal{T} = [T_1, T_2, ...T_K]$. In this way, the state of a DNN model during multitask training can be described by a *state vector* $\mathbf{S} \in \mathbb{R}^K$.

Without loss of generality, we define the range of each probing result in $\mathbb{R}$. Empirically, many scores are used as probing results, including correlation scores (Gupta et al., 2015), usable $\mathcal{V}$-information (Pimentel and Cotterell, 2021), minimum description length (Voita and Titov, 2020), and combinations thereof (Hewitt and Liang, 2019). Currently, the most popular probing results are written in accuracy values, ranging from 0 to 1. We do not require the actual choice of probing metric as long as all probing tasks adopt the same metric.

**From model state to dataset state**   Now that we have the vectorized readout values of the states,

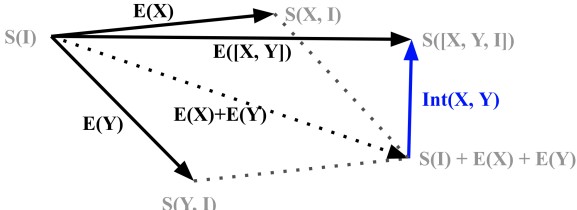

Figure 2: Schematic illustration of the dataset effects. $\mathbf{E}(\cdot)$ are the individual effects, $\text{Int}(\mathbf{X}, \mathbf{Y})$ is the interaction effect.

how is the state $S$ determined? In general, there are three factors: the previous state, the dataset, and the training procedure applied to the model since the previous state. We can factor out the effects of the previous state and the training procedure.

To factor out the effect of the previous state, we introduce the concept of an "initial state". The initial state is specified by the structures and the parameters of the DNNs. If we study the effects of data in multitask fine-tuning BERT, then the initial state is the BERT model before any fine-tuning. Let us write the initial state as $S_0$. Based on this initial state, the dataset $X$ specifies a training task that leads the model from the initial state $S_0$ to the current state $S_X$.

To factor out the effect of the training procedure, we assume the dataset $X$, together with the training objective, defines a unique global optimum. Additionally, we consider the training procedure can eventually reach this optimum, which we write as the *dataset state*, $S_X$. Practically, various factors in the training procedures and the randomness in the sampled data may lead the models towards some local optima, but the framework considers the global optimum when specifying the dataset state.

## 4 Dataset effects

### 4.1 Individual dataset effect

We define the effect of a dataset $x$ as:

$$\mathbf{E}(X) \equiv \mathbf{S}([X, I]) - \mathbf{S}(I) \in \mathbb{R}^K \qquad (1)$$

Here $[X, I]$ denotes combining datasets $X$ and $I$, and $\mathbf{S}([X, I])$ and $\mathbf{S}(I)$ are the state vectors of the states $S_{[X,I]}$ and $S_I$ respectively. $\mathbf{E}(X)$ describes how does the linguistic ability of the model shift along the $K$ dimensions that we probe. In other words, $\mathbf{E}(X)$ describes the multi-dimensional fine-tuning effect of the dataset $X$.

*Remark 1*: The effect of $X$ depends on a "reference state" $S_I$ where $I$ is used to describe the

dataset leading to the reference state $S_I$. In an edge case, the dataset $X$ is a subset of the dataset $I$, so $S_{[X,I]} = S_I$. This can be attributed to the fact that our definition $S_I$ is the global optimum of a model fine-tuned on $I$ among all possible training procedures, including re-sampling. Hence, the dataset $X$ should have no effect on the model, which can be verified by $\mathbf{E}(X) = \mathbf{S}([X, I]) - \mathbf{S}(I) = \mathbf{0}$.

*Remark 2*: In another scenario, $X$ consists of $I$ plus only one other data point $z$. Then $\mathbf{E}(X)$ degenerates to the effect of the data point $z$.

*Remark 3*: We assume $X$ does not overlap with $I$ in the rest of this paper. This assumption stands without loss of generality since we can always redefine $X$ as the non-overlapping data subset.

*Remark 4*: The dataset effects form an Abelian group – Appendix §A.3 contains the details.

### 4.2 Interaction effect

**Motivating example** Let us first consider an example of detecting the sentiment polarity. Suppose three abilities can contribute to addressing the sentiment polarity task:

A1: Parsing the structure of the review.
A2: Recognizing the tense of the review.
A3: Detecting some affective keywords such as "good".

Consider two sentiment polarity datasets, $X$ and $Y$. In $X$, all positive reviews follow a unique syntax structure where all negative reviews do not. In $Y$, all positive reviews are written in the present tense where all negative reviews in the past tense. The problem specified by dataset $X$ can be solved by relying on both A1 and A3, and the problem specified by dataset $Y$ can be solved by relying on both A2 and A3. Imagine a scenario where after training on both $X$ and $Y$, a model relies solely on A3 to predict the sentiment polarity. This behavior is caused by the interaction between $X$ and $Y$. Using the terms of our state-vector framework, there is an *interaction effect* between $X$ and $Y$ with a positive value along the dimension of A3, and a negative value along the dimensions of A1 and A2.

**Definition of the interaction effect** Let us define the *interaction effect* between two datasets, $X$ and $Y$ as:

$$\text{Int}(X, Y) = \mathbf{E}([X, Y]) - (\mathbf{E}(X) + \mathbf{E}(Y)) \quad (2)$$

This is the difference between the dataset effect of $[X, Y]$ ($X$ and $Y$ combined), and the sum of

the effects of $X$ and $Y$ (as if $X$ and $Y$ have no interactions at all).

*Remark 1:* An equivalent formulation is:

$$\text{Int}(X,Y) = \mathbf{S}([X,Y,I]) - \mathbf{S}([X,I]) \\ - \mathbf{S}([Y,I]) + \mathbf{S}(I) \quad (3)$$

*Remark 2:* What if $X$ and $Y$ share some common items? This would introduce an additional effect when merging $X$ and $Y$. An extreme example is when $X$ and $Y$ are the same datasets, where $[X,Y]$ collapses to $X$. Then $\text{Int}(X,Y) = -\mathbf{E}(X)$ where it should be $\mathbf{0}$. The reason for this counter-intuitive observation is that the "collapse" step itself constitutes a significant interaction. Our "datasets do not overlap" assumption avoids this problem. This assumption goes without loss of generality because we can always redefine $X$ and $Y$ to contain distinct data points.

**A linear regression view** The interaction effect as defined in Eq. 2, equals the $\beta_3$ coefficient computed by regressing for the state vector along each of the $K$ dimensions:

$$\mathbf{S}^{(k)} = \beta_0^{(k)} + \beta_1^{(k)} i_x + \beta_2^{(k)} i_y + \beta_3^{(k)} i_x i_y + \epsilon \quad (4)$$

where $i_x$ and $i_y$ are indicator variables, and $\epsilon$ is the residual. If dataset $X$ appears, $i_x = 1$, and $i_x = 0$ otherwise. The same applies to $i_y$. $\mathbf{S}^{(k)}$ is the $k^{th}$ dimension in the state vector. The correspondence between the indicator variables and $\mathbf{S}$ as listed in Table 1:

| $i_x$ | $i_y$ | $\mathbf{S}$ |
|---|---|---|
| 0 | 0 | $\mathbf{S}(I)$ |
| 1 | 0 | $\mathbf{S}([X,I])$ |
| 0 | 1 | $\mathbf{S}([Y,I])$ |
| 1 | 1 | $\mathbf{S}([X,Y,I])$ |

Table 1: Correspondence between the indicator variables and the regression targets.

Please refer to Appendix A.4 for a derivation of the equivalence. The Eq. 4 formulation allows us to apply an ANOVA, which allows us to decide if the interaction effect is significant (i.e., the $p$-value of $\beta_3$ is smaller than the significance threshold).

## 5  Experiments

### 5.1  Models

Experiments are conducted using two popular language models: BERT-base-cased (Devlin et al.,

2019) and RoBERTa-base (Liu et al., 2019). Doing this allows us to compare results and determine whether dataset effects hold despite model choice.

### 5.2  Fine-tuning

The GLUE benchmark (Wang et al., 2018) consists of 3 types of natural language understanding tasks: single-sentence, similarity and paraphrase, and inference tasks. Two tasks from each category were selected to fine-tune models.

**Single-sentence tasks** COLA (Warstadt et al., 2018) labels whether a sentence is a grammatical English sentence or not. SST2 (Socher et al., 2013) labels whether a sentence has positive or negative sentiment.

**Similarity and paraphrase tasks** MRPC (Dolan and Brockett, 2005) labels whether sentences in a pair are semantically equivalent or not. STSB (Cer et al., 2017) accomplishes the same objective except it provides a similarity score between 1-5.

**Inference tasks** QNLI (Rajpurkar et al., 2016) takes a sentence-question pair as input and labels whether the sentence contains the answer to the question. RTE (Dagan et al., 2005; Haim et al., 2006; Giampiccolo et al., 2007; Bentivogli et al., 2009) labels whether a given conclusion is implied from some given text.

The multitask model resembles that of Radford et al. (2018). It consists of a shared encoder with a separate classification layer per task – Figure 4 shows an illustration. This was made possible by leveraging HuggingFace's Transformers library (Wolf et al., 2020). Model hyperparameters are the same as those listed in Table 3 of Mosbach et al. (2020), except 3 epochs were used for all experiments. This is valid, as their experiments incorporated both BERT-base-cased and RoBERTa-base models. Model checkpoints were saved every 6 optimization steps, with the final model being the one with the lowest training loss.[1]

To mitigate the effect of train set size per task, train datasets were reduced to 2,490 examples per task, which corresponds to the size of the smallest train dataset (COLA). GLUE benchmarks such as QQP, MNLI, and WNLI (Levesque et al., 2012) were excluded because their train sets were too large (more than 300K examples) or too small (fewer than 1K examples).

---

[1]Downstream task performance can be accessed in our repository.

## 5.3 Probing

We use the SentEval suite (Conneau and Kiela, 2018) to build proxies for the idealized probes that vectorize the model states. SentEval contains the following tasks:

- **Length:** given a sentence, predict what range its length falls within (0: 5-8, 1: 9-12, 2: 13-16, 3: 17-20, 4: 21-25, 5: 26-28).
- **WC:** given a sentence, predict which word it contains from a target set of 1,000 words.
- **Depth:** given a sentence, predict the maximum depth of its syntactic tree.
- **TopConst:** given a sentence, predict its constituent sequence (e.g. NP_VP_.: noun phrase followed by verb phrase).
- **BigramShift:** given a sentence, predict whether any two consecutive tokens of the original sentence have been inverted.
- **Tense:** given a sentence, predict whether its main verb is in the past or present tense.
- **SubjNumber:** given a sentence, predict whether the subject of its main clause is singular or plural.
- **ObjNumber:** given a sentence, predict whether the direct object of its main clause is singular or plural.
- **OddManOut:** given a sentence, predict whether any verb or noun of the original sentence was replaced with another form with the same part of speech.
- **CoordInv:** given a sentence, predict whether two coordinated casual conjoints of the original sentence have been inverted.

For each probing task, we downsample the datasets to 10% of their original sizes or 12K samples per task (10K train, 1K validation, 1K test). This is valid, as datasets of similar sizes usually have sufficient statistical power (Card et al., 2020; Zhu et al., 2022b). WC was removed from consideration, as its performance would have been significantly compromised given that it possesses 1000 ground truths. We built our training pipeline based on the SentEval library and used the default config.[2] The default architecture is a single classification layer on top of the frozen fine-tuned encoder.

## 5.4 Multitask settings

As the number of tasks increases, the number of combinations increases exponentially. To compute

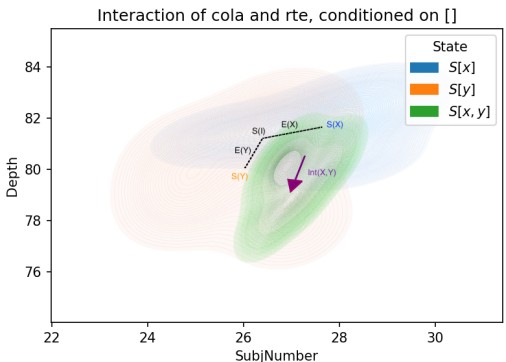

Figure 3: An example of interaction effect between x=COLA and y=RTE, of RoBERTa. The purple arrow visualizes the interaction effect. This figure plots two linguistic ability dimensions – syntax tree depth and subject number.

the dataset states in a realistic time, we group the experiments by the format of the tasks (e.g., single-sentence, similarity, inference) and impose selection conditions, reducing the total number of fine-tuning experiments from around 10k to 300 per model. Note that 300 fine-tuning experiments per model are still nontrivial, but they can be completed within two months. Section A.2 in Appendix contains the details.

## 6 Results

### 6.1 Individual dataset effect

Some observations regarding individual dataset effects are noted below. Tables 2 and 3 summarize average individual GLUE dataset effects for BERT and RoBERTa models, respectively. Tables 4 and 9 break down the average individual effects of COLA and SST2, respectively.[3]

**Model choice** Per Tables 2 and 3, there is no consistent agreement between significant dimensions (i.e., dimensions marked as *, **, or ***). In fact, of the combined 33 significant individual effects observed in Tables 2 and 3, only 13 ($\approx 40\%$) can be confirmed by both BERT and RoBERTa models. This demonstrates that datasets can have different effects depending on model architecture.

**Dataset composition** MRPC and STSB accomplish very similar tasks, but impact different dimensions. Matching significant dimensions for MRPC and STSB amount to 2 of 7 for BERT (see Table 2) and 1 of 5 for RoBERTa (see Table 3).

---

[2]SentEval code and default config are accessible here.

[3]Tables for other datasets can be accessed in our repository.

| Dataset | Model | Length | Depth | TopConst | BigramShift | Tense | SubjNumber | ObjNumber | OddManOut | CoordInv |
|---------|-------|--------|-------|----------|-------------|-------|------------|-----------|-----------|----------|
| COLA | BERT | −0.36 | −0.47 | −0.83 | 6.35*** | −0.23 | −0.38 | −0.69* | 1.93*** | 1.09*** |
| SST2 | BERT | −4.08*** | −0.51 | −3.92*** | −1.99*** | −0.24 | −1.75*** | −1.73*** | −0.79* | 0.06 |
| MRPC | BERT | 0.65 | 0.75*** | −2.34*** | 0.55 | −0.45*** | −0.78* | 1.07*** | 1.13*** | 1.34*** |
| STSB | BERT | −0.53 | 0.27 | −1.66* | −0.24 | −0.3* | 0.36 | 0.2 | 0.36 | −0.09 |
| QNLI | BERT | 0.05 | 0.33 | 0.85 | −0.49 | −0.16 | 0.81* | 0.16 | −0.34 | 1.33*** |
| RTE | BERT | 0.83 | 0.34 | −0.58 | −0.24 | −0.21 | −0.15 | −0.65* | 0.16 | 0.64 |

Table 2: Individual effects of GLUE datasets, on BERT. *, **, and *** stand for $p < 0.05$, $p < 0.01$, and $p < 0.001$, respectively, using two-sample $t$-test with $dof = 8$.

| Dataset | Model | Length | Depth | TopConst | BigramShift | Tense | SubjNumber | ObjNumber | OddManOut | CoordInv |
|---------|-------|--------|-------|----------|-------------|-------|------------|-----------|-----------|----------|
| COLA | RoBERTa | −2.75*** | 0.28 | 2.17* | 9.17*** | 1.21*** | 0.26 | 0.59 | 3.66*** | 2.0*** |
| SST2 | RoBERTa | −7.12*** | −1.86*** | −5.88*** | −4.43*** | −1.57*** | −3.92*** | −3.96*** | −1.85*** | −2.08*** |
| MRPC | RoBERTa | −0.06 | 0.25 | −0.87 | −1.68 | −1.01* | −0.61 | −0.45 | 0.07 | 1.0 |
| STSB | RoBERTa | 0.03 | 0.14 | −3.17*** | −3.0* | −1.84*** | −3.11*** | −1.65* | −0.43 | 0.16 |
| QNLI | RoBERTa | −0.67 | 0.04 | −0.13 | −0.57 | −0.56 | −0.33 | −1.47* | 0.5 | 0.13 |
| RTE | RoBERTa | 0.78 | −0.84* | −0.58 | −0.32 | −0.81 | −0.85 | −2.04*** | −0.25 | 0.0 |

Table 3: Individual effects of GLUE datasets, on RoBERTa. *, **, and *** stand for $p < 0.05$, $p < 0.01$, and $p < 0.001$, respectively, using two-sample $t$-tests with $dof = 8$.

Although they both are paraphrasing tasks, their samples are extracted from different sources and have different ground truths (i.e., MRPC is a binary task, STSB is an ordinal task). Hence, the composition of datasets can affect what individual effects they will contribute.

**Dataset type** The inference datasets (QNLI, RTE) do not have much significant impact on the set of probing dimensions. In all cases, both datasets have no more than two significant dimensions (see Tables 2 and 3). This is far fewer than single-sentence tasks (COLA, SST2), whose effects span many more dimensions (see Tables 2 and 3). We hypothesize that this can be attributed to the fact that SentEval probing tasks assess linguistic information captured on the sentence level. By fine-tuning on single-sentence datasets, models are more likely to learn these relevant sentence properties and incorporate them in their embeddings. Inference datasets are more complex and require models to learn linguistic properties beyond the sentence scope, such as semantic relationships between sentences. Similarity and paraphrase datasets fall in between single-sentence and inference datasets with respect to complexity and linguistic scope, which explains why MPRC and STSB impact more dimensions than QNLI and RTE but fewer dimensions than COLA and SST2.

**Reference state** In most cases, significant dataset dimensions varied with different reference states. From Table 4, it is clear that significant individual effects of COLA are inconsistent between experiments, besides BigramShift (for both models) and OddManOut (positive effect for RoBERTa only). The same conclusion is valid for datasets other than COLA. This implies that there are inherent interaction effects between datasets that also influence results. Note that if we add a dataset to a large number of varying reference states and observe that there are persistent, significant dimensions across these experiments, then this is a strong empirical indication of a dataset's effect on this set of dimensions (e.g., see COLA's effect on BigramShift in Table 4 and SST2's effect on Length in Table 9). In the case of our experiments, for argument's sake, we impose that a dataset effect must appear in at least 70% reference states. This lower bound can be adjusted to 60% or 80%, but wouldn't result in any major adjustments as dataset effects tend to appear in fewer than 50% of our experiments. Table 5 summarizes such instances and supports previous statements made regarding model choice (some effects are only observed on RoBERTa) and dataset type (both datasets are single-sentence tasks). The low number of table entries further justifies that there are other confounding variables, including but not limited to model choice, hyperparameter selection, and dataset interactions.

**Spill-over** We observed that some syntactic datasets have effects along the semantic dimensions and vice-versa. This is odd, as learning sentiment shouldn't be correlated to e.g., losing the ability to identify the syntactic constituents or swapped word order. We refer to such effects as "spill-over" ef-

| Dataset | Reference | Model | Length | Depth | TopConst | BigramShift | Tense | SubjNumber | ObjNumber | OddManOut | CoordInv |
|---|---|---|---|---|---|---|---|---|---|---|---|
| COLA | I | BERT | −1.32 | 0.8 | −2.94* | 6.32*** | −1.14*** | −0.02 | −0.34 | 3.56*** | 0.3 |
| COLA | SST2 | BERT | −1.16 | −1.2 | −0.0 | 7.4*** | −0.7 | 0.22 | −0.7 | 2.22*** | 1.82 |
| COLA | MRPC | BERT | −0.6 | 0.78 | −0.06 | 5.6*** | 0.44 | 0.14 | −0.18 | 0.56 | 0.92 |
| COLA | STSB | BERT | −2.84 | −1.0 | −0.8 | 6.48*** | 0.94 | 0.08 | −1.94* | 3.34* | 3.54*** |
| COLA | QNLI | BERT | 1.24 | −1.26 | −1.26 | 7.2*** | −0.24 | −0.96 | −0.66 | 2.46 | 0.58 |
| COLA | RTE | BERT | 1.5 | −1.82* | −3.3* | 5.94*** | −0.26 | −2.02 | −0.68 | 2.16 | 0.96 |
| COLA | MRPC QNLI | BERT | 0.62 | 0.46 | −2.86 | 5.22*** | −1.08* | −0.62 | −0.46 | 1.12 | 0.28 |
| COLA | MRPC RTE | BERT | −0.58 | −0.64 | 0.92 | 5.52*** | −0.32 | 1.34 | −0.24 | 1.46*** | 1.02 |
| COLA | STSB QNLI | BERT | −0.32 | −0.52 | 1.12 | 6.56*** | 0.24 | −1.26 | −0.44 | 1.38* | 0.38 |
| COLA | STSB RTE | BERT | −0.18 | −0.26 | 0.92 | 7.24*** | −0.2 | −0.7 | −1.22 | 1.08 | 1.08 |
| COLA | I | RoBERTa | 2.18*** | 1.24 | 8.62*** | 5.78*** | 0.28 | 0.44 | −1.78* | 4.64*** | 2.7* |
| COLA | SST2 | RoBERTa | −0.66 | 2.18 | 6.18*** | 13.08*** | 2.44 | 3.28 | 2.42 | 4.4*** | 4.52* |
| COLA | MRPC | RoBERTa | −5.48* | −0.34 | −1.58 | 7.94*** | 0.14 | 0.16 | 0.78 | 2.62* | 1.6 |
| COLA | STSB | RoBERTa | −3.84* | −1.0 | 3.8 | 11.9*** | 3.46* | 0.98 | −0.1 | 4.6* | 2.2 |
| COLA | QNLI | RoBERTa | −3.68 | −0.5 | 0.38 | 5.86*** | −0.1 | −1.88 | −0.96 | 3.84*** | 3.02* |
| COLA | RTE | RoBERTa | −1.82 | 1.04 | 6.04* | 7.78*** | 0.36 | −0.5 | 0.28 | 4.18*** | 4.28*** |
| COLA | MRPC QNLI | RoBERTa | −5.58* | −0.7 | 0.24 | 8.8*** | 0.78 | 1.0 | 0.86 | 2.18* | 0.54 |
| COLA | MRPC RTE | RoBERTa | 0.56 | 1.34* | 3.1 | 8.64*** | 3.32* | 1.54 | 2.54 | 3.3* | 0.62 |
| COLA | STSB QNLI | RoBERTa | −4.04 | 0.38 | 0.88 | 11.02*** | 1.22 | 0.28 | 1.28 | 3.4*** | 0.1 |
| COLA | STSB RTE | RoBERTa | −5.12*** | −0.8 | −5.98 | 10.86*** | 0.22 | −2.74 | 0.62 | 3.46 | 0.42 |

Table 4: Individual effects of COLA dataset with different reference states. *, **, and *** stand for $p < 0.05$, $p < 0.01$, and $p < 0.001$, respectively, computed using two-sample $t$-test with $dof = 8$.

| Dataset | Dimension | Effect | Model(s) |
|---|---|---|---|
| COLA | BigramShift | + | Both |
| COLA | OddManOut | + | RoBERTa |
| SST2 | Length | - | Both |
| SST2 | TopConst | - | RoBERTa |
| SST2 | BigramShift | - | Both |

Table 5: Individual effects that are observed in at least 70% of reference states.

fects. For instance, Tables 2 and 3 suggest that fine-tuning on COLA (a syntactic dataset) has a positive effect on OddManOut and CoordInv (semantic dimensions). This is unexpected, given OddManOut and CoordInv probing datasets consist of grammatically acceptable sentences, they only violate semantic rules. Conversely, fine-tuning on SST2 (a semantic dataset) hurts TopConst and BigramShift (syntactic or surface-form dimensions). We hypothesize that the spill-over individual effects are likely due to the aforementioned confounding variables inducing false correlations. More rigorous analysis is required to identify these variables and their effects.

## 6.2 Interaction effect

The average interaction effects of datasets on RoBERTa and BERT are listed in Table 6 and Table 7, respectively.

**Interaction effects are concentrated** The interaction effects are not always significant along the probing dimensions. More specifically, in most (28

out of 30) scenarios listed in Tables 6 and Table 7, the significant interactions concentrate on no more than three dimensions. The remaining two scenarios are both SST2 and RTE (on RoBERTa and BERT, respectively).

**Interaction effects can occur with insignificant individual effects** Many interaction effects are observed along the linguistic dimensions where neither of the participating datasets has a significant individual effect. For example, all significant interactions along the "Depth" dimension for BERT have this characteristic. Apparently, even if a dataset does not have a significant individual effect along a dimension, it could still interact with the other dataset along this dimension.

**Interaction effects are characteristic** The significant dimensions differ across the datasets. Even the datasets with similar purposes do not demonstrate identical interaction effects with the same "third-party" dataset. For example, both MRPC and STSB target at detecting paraphrasing. When interacting with COLA, STSB has an insignificant effect on the TopConst dimension, while MRPC has a significant negative effect along the same dimension. Can we predict the dimensions of the significant effects, just by analyzing the datasets? Rigorous, systematic studies in the future are necessary to give deterministic answers.

**Similar datasets interact less** The interactions of similar datasets appear to interact less significantly than those "less similar" datasets. Among

| $X$ | $Y$ | Model | Dataset effect dimensions | | | | | | | | |
|---|---|---|---|---|---|---|---|---|---|---|---|
| | | | Length | Depth | TopConst | BigramShift | Tense | SubjNumber | ObjNumber | OddManOut | CoordInv |
| COLA | SST2 | RoBERTa | −2.84 | 0.94 | −2.44 | 7.30*** | 2.16 | 2.84 | 4.20* | −0.24 | 1.82 |
| COLA | MRPC | RoBERTa | −7.66*** | −1.58 | −10.20*** | 2.16** | −0.14 | −0.28 | 2.56 | −2.02 | −1.10 |
| COLA | STSB | RoBERTa | −6.02** | −2.24* | −4.82 | 6.12*** | 3.18** | 0.54 | 1.68 | −0.04 | −0.50 |
| COLA | QNLI | RoBERTa | −5.86** | −1.74* | −8.24*** | 0.08 | −0.38 | −2.32 | 0.82 | −0.80 | 0.32 |
| COLA | RTE | RoBERTa | −4.00* | −0.20 | −2.58 | 2.00** | 0.08 | −0.94 | 2.06 | −0.46 | 1.58 |
| SST2 | MRPC | RoBERTa | −3.74* | 0.96 | −2.72 | 5.64* | 1.98 | 4.40* | 3.74* | 0.88 | −0.50 |
| SST2 | STSB | RoBERTa | −2.86 | 0.06 | 0.36 | 6.64*** | 4.40* | 4.20 | 3.60* | 1.44 | 1.44 |
| SST2 | QNLI | RoBERTa | 0.18 | 1.02 | 0.18 | 2.12 | 2.64 | 3.08 | 3.84 | 0.60 | 2.82* |
| SST2 | RTE | RoBERTa | −0.20 | 1.48 | 8.14** | 5.78*** | 4.68** | 5.56** | 5.48** | 0.50 | 5.96*** |
| MRPC | STSB | RoBERTa | −3.34 | −3.14** | −1.08 | 3.84* | 3.58** | 2.98 | 0.50 | −1.92 | −0.78 |
| MRPC | QNLI | RoBERTa | −2.04 | −1.14 | −4.44 | 0.64 | 0.42 | 0.32 | 1.44 | 0.90 | 1.06 |
| MRPC | RTE | RoBERTa | −6.58** | −1.58* | −3.10 | 1.88 | −1.48 | 0.78 | 0.04 | −0.10 | 1.72 |
| STSB | QNLI | RoBERTa | 0.12 | −2.42* | −3.10 | 1.18 | 2.78 | 0.54 | 0.78 | 0.14 | 1.16 |
| STSB | RTE | RoBERTa | 1.12 | −1.42 | 6.28* | 3.28* | 3.12 | 1.52 | 1.84 | 0.26 | 2.00 |
| QNLI | RTE | RoBERTa | −0.10 | −0.74 | −0.68 | 0.96 | 0.06 | −0.76 | 1.30 | 0.40 | 1.86 |

Table 6: Interaction effects between GLUE datasets, on RoBERTa. *, ** and *** stand for $p < 0.05$, $p < 0.01$, and $p < 0.001$, respectively, from the ANOVA test of the interaction effect.

the 30 scenarios in Tables 6 and 7, only two scenarios show no significant interactions along any dimensions: (MRPC, QNLI) and (QNLI, RTE), both on RoBERTa, and both involve strong similarities between the datasets: QNLI and RTE test the same downstream task (infer the textual entailment), and MRPC and QNLI both involve an intricate understanding of the semantics of the text.

## 7 Discussion

**Checking the effects before using datasets**  Considering that the datasets can show spill-over effects when used independently and interaction effects when used jointly, we call for more careful scrutiny of datasets. While the model developers already have a busy working pipeline, we call for model developers to at least be aware of spill-over effects of the datasets. Considering the possibly negative effects to the models' linguistic abilities, adding datasets to the model's training might not always be beneficial.

**Documentation for datasets**  The transparency of model development pipelines can be improved, and better documentation of the data is a crucial improvement area (Gebru et al., 2021; Paullada et al., 2021). Recently, Pushkarna et al. (2022) described some principles for unified documentation of datasets: flexible, modular, extensible, accessible, and content-agnostic. The dataset effect can be a module in the dataset documentation. In addition to documenting the basic properties of the datasets, it would be great to also note how the dataset has potential "spill-over" effects and "interaction effects". This is better done via a joint effort from

the AI community.

**From data difficulty to "dataset effects"**  While the difficulty of the dataset is a uni-dimensional score, the effect of datasets can be multi-dimensional. Improving the difficulty of datasets (e.g., by identifying adversarial examples and challenging datasets) has been shown to improve performance (Ribeiro et al., 2020; Gardner et al., 2020). The consideration of multi-dimensional dataset effects can potentially introduce similar benefits.

## 8 Conclusion

We propose a state-vector framework to study *dataset effects*. The framework uses probing classifiers to describe the effects of datasets on the resulting models along multiple linguistic ability dimensions. This framework allows us to identify the individual effects and the interaction effects of a number of datasets. With extensive experiments, we find that the dataset effects are concentrated and characteristic. Additionally, we discuss how the state-vector framework to study dataset effects can improve the dataset curation practice and responsible model development workflow.

## 9 Limitations

**Probing tests may not be idealized**  When formulating the framework, we consider idealized probes – 100% valid and reliable. In reality, probing tests are unfortunately not ideal yet. We follow the common practice of setting up the probing classifiers to allow fair comparison with their literature.

| $X$ | $Y$ | Model | Dataset effect dimensions | | | | | | | | |
|-----|-----|-------|--------|-------|----------|-------------|-------|------------|-----------|-----------|----------|
| | | | Length | Depth | TopConst | BigramShift | Tense | SubjNumber | ObjNumber | OddManOut | CoordInv |
| COLA | SST2 | BERT | 0.16 | −2.00* | 2.94 | 1.08* | 0.44 | 0.24 | −0.36 | −1.34 | 1.52 |
| COLA | MRPC | BERT | 0.72 | −0.02 | 2.88 | −0.72 | 1.58* | 0.16 | 0.16 | −3.00** | 0.62 |
| COLA | STSB | BERT | −1.52 | −1.80 | 2.14 | 0.16 | 2.08** | 0.10 | −1.60 | −0.22 | 3.24* |
| COLA | QNLI | BERT | 2.56 | −2.06* | 1.68 | 0.88 | 0.90* | −0.94 | −0.32 | −1.10 | 0.28 |
| COLA | RTE | BERT | 2.82 | −2.62** | −0.36 | −0.38 | 0.88 | −2.00 | −0.34 | −1.40 | 0.66 |
| SST2 | MRPC | BERT | 0.66 | −0.32 | 2.74 | −0.56 | 1.08 | 0.60 | −1.62 | −2.10* | 0.16 |
| SST2 | STSB | BERT | −0.70 | −0.90 | 1.50 | −0.82 | 1.12 | −1.56 | −3.66*** | −0.52 | 2.80* |
| SST2 | QNLI | BERT | 0.36 | −0.58 | 0.06 | 0.46 | 0.68 | −2.30 | −2.80* | −3.04** | −1.28 |
| SST2 | RTE | BERT | 1.82 | −2.26* | −2.02 | −0.62 | 0.64 | −2.96* | −2.00* | −2.60* | −1.30 |
| MRPC | STSB | BERT | −2.00 | −0.48 | 5.48** | −0.94 | 2.22* | 2.42* | −2.36 | −0.12 | 3.42** |
| MRPC | QNLI | BERT | −0.28 | −1.10 | 4.42* | 0.50 | 1.76* | 1.04 | −0.94 | −1.50 | −0.36 |
| MRPC | RTE | BERT | 2.38 | −0.48 | 0.34 | −0.54 | 1.42* | −2.38* | −2.34* | −2.50* | −0.32 |
| STSB | QNLI | BERT | −0.48 | −0.58 | 1.28 | 0.46 | 1.44* | 0.14 | −3.18** | 0.96 | 1.70* |
| STSB | RTE | BERT | 1.06 | −1.88* | 0.30 | −0.82 | 1.70* | −1.60 | −3.68*** | −0.04 | 2.20 |
| QNLI | RTE | BERT | 3.46 | −1.32 | −2.12 | 0.12 | 0.36 | −3.38 | −2.76** | 0.70 | −1.20 |

Table 7: Interaction effects between GLUE datasets, on BERT. *, ** and *** stand for $p < 0.05$, $p < 0.01$, and $p < 0.001$, respectively, for the ANOVA test of the interaction effect.

We run the probing experiments on multiple random seeds to reduce the impacts of randomness.

**Model training may not be optimal** Empirically, the datasets included in our analyses are limited to the fine-tuning stage. Previous work found distinct "stages" during the training of DNNs where the DNNs respond to the data samples differently. For example, Shwartz-Ziv and Tishby (2017) referred to the stages as "drift phase" and the "diffusion phase". The means of the gradients are drastically different between the two stages. Tänzer et al. (2022) identified a "second stage" where the models do not overfit to noisy data labels. In the framework of this paper, we consider the *ideal* model training, where our states are defined as the global optimum where the model arrives.

**Interaction effects of more than two tasks** The interaction effect is defined between two tasks. We believe this framework can generalize to more than two tasks, but the empirical verification is left to future work.

**Coverage of experiments** As the number of datasets we consider increases, the number of experiments in total grows exponentially. It is unrealistic to go through the set of all combinations in our experiments, so we picked some experiments and organized them given the categories of the desired effects (instead of the observed effects) of the datasets. Additional experiments that test the exact interaction effects are left to future works. Also, we only considered classification-type tasks in the experiments. While this state-vector framework naturally generalizes to other tasks, including cloze and next-sentence prediction, the empirical observations are left to future works as well. We consider the fine-tuning setting in the experiments. Another setting, language model pre-training, also involves classification-type tasks and usually has larger sets of labels. Our theoretical framework generalizes to the pre-training setting as well.

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

.

# A    Appendix

## A.1    Figure illustrating the experimental setup

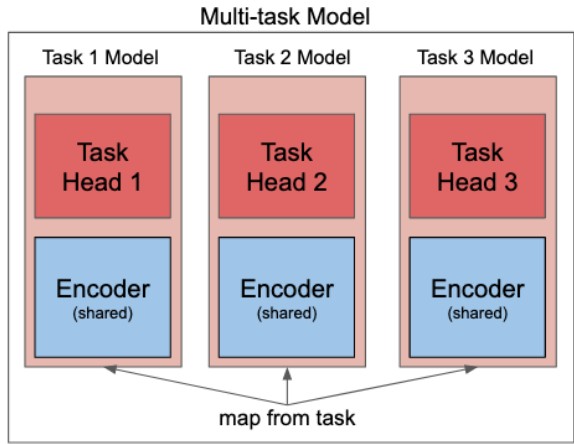

Figure 4: Multitask model architecture for fine-tuning experiments.[4]

## A.2    Additional details on multitask settings

Consider the number of different multitask fine-tuning settings possible assuming a constant random seed. If we fine-tune encoders task-by-task (e.g., first COLA, then MRPC) such that order matters, then this problem is equivalent to the number of ordered subsets that can be formed from the set $T = \{\text{COLA, SST2, MRPC, STSB, QNLI, RTE}\}$. This computes to $\sum_{n=0}^{6} \frac{6!}{(6-n)!} = 1957$ models per encoder, or $1957 \cdot 2 = 3914$ total models. Note the lower bound of the summation is 0, as the empty set corresponds to the baseline model (i.e., no fine-tuning).

If we discount ordering effects, then the problem reduces to the number of subsets $T$ contains. This evaluates to $2^6 = 64$ models per encoder or $64 \cdot 2 = 128$ total models, which results in far fewer experiments. Ordering effects can be discounted by aggregating training samples of each task, then randomly sampling from this combined dataset when training. Each sample will have a tag indicating which task it pertains to, so it can be redirected to the correct classification head during training.

Note that we initially assume a constant random seed. Later, we expanded to five random seeds (42, 1, 1234, 123, 10) to allow statistical significance testing. This increases total models to $128 \cdot 5 = 640$, which requires excessive compute resources. There is also the need to organize experiments better to

---

[4]Experimental setup based on this guide

| Marker | N. Groups | N. Tasks | N. Experiments |
|--------|-----------|----------|----------------|
| $I$ | 0 | 0 | 1 |
| $A$ | 1 | 1 | 6 |
| $B$ | 1 | 2 | 3 |
| $C$ | 2 | 1 | 12 |
| $D$ | 3 | 1 | 8 |
| $E$ | 2 | 2 | 3 |
| $F$ | 3 | 2 | 1 |

Table 8: Multitask states with counts of groups, tasks per group, and experiments per encoder.

illustrate potential individual and interaction effects clearly.

To address these issues, we impose the following condition: **experiments must constitute of equal task counts per task group OR all tasks must belong to the same task group.** Recall the task groups from Section 5.2 to be single-sentence, similarity and paraphrase, and inference. This enables us to organize the experiments by marking the states as follows:

- $I$: The initial state.
- $A$: The model is trained on one dataset.
- $B$: The model is trained on two datasets from the same group.
- $C$: The model is trained on two datasets from different groups.
- $D$: The model is trained on three datasets from different groups.
- $E$: The model is trained on four tasks from two groups (two per group).
- $F$: The model is trained on six tasks from three groups (two per group)

As demonstrated in Table 8, the total number of models we need to train is reduced to $34 \cdot 2 \cdot 5 = 340$. Designing experiments this way allows framing the dataset effects.

The individual effects can be framed as transitions between marked states (i.e. adding some task to one state yields another state). For example, $I \rightarrow A$ can reflect the individual effect of a dataset, conditioned on the "no-fine-tuning initial state" $I$. $A \rightarrow B$ denotes the individual effect of a dataset $X$, conditioned on the initial state that contains a dataset ($Y$), where $Y$ is in the same group as $X$.

The interaction effects can be framed as combinations between multiple states. For example, two states labeled $A$ (with dataset $X$ and $Y$, respectively) and a state labeled $B$ (with datasets $[X, Y]$) can jointly define the interaction between the datasets $X$ and $Y$. This can be written as $B = A + A$.

This labeling mechanism of the states can support the following effects:

- Individual effects: $I \to A$, $A \to B$, $A \to C$, $C \to D$
- Interaction effects: $B = A + A$, $C = A + A$, $D = A + C$, $E = B + B$, $E = C + C$, $F = B + E$, $F = D + D$

Note that although it is possible to compute interaction effects of more than two datasets, we chose not to focus on these cases as it adds an extra layer of complexity. Hence, we only consider the following interaction effects: $B = A + A$, $C = A + A$. This means we don't need to train models for states $E$ and $F$, reducing total experiments to $30 \cdot 2 \cdot 5 = 300$.

### A.3 Additional math motivation

Here we provide some additional mathematical motivations for the proposed state-vector framework: dataset effects form an Abelian group.

Given a reference state $S_I$, the collection of all possible dataset effects $\mathbf{E}(X) \in \mathcal{E}$ forms an additive Abelian group. Here we show that $\mathcal{E}$ satisfies the requirements.

*Existence of zero.* We already know that the identity element is $\mathbf{0} \in \mathbb{R}^K$. Intuitively, the identity element corresponds to "no effect" for this dataset.

*Existence and closure of addition.* The addition operation refers to the vector addition. Since $\mathcal{E}$ is defined under $\mathbb{R}^K$, it is closed under addition. Note that due to the interaction effect, addition does *not* refer to applying two datasets together to the "bucket" of data for multitask training.

*Existence and closure of negation.* The negation operation refers to flipping the direction of a vector in $\mathcal{E}$. Empirically, negating $\mathbf{E}(X)$ involves a counterfactual query of the effect of a dataset: if $X$ were not applied, what would have been the effect on the state of the model?

*Associativity and commutativity.* Vector addition satisfies associativity: $(\mathbf{E}(X) + \mathbf{E}(Y)) + \mathbf{E}(Z) = \mathbf{E}(X) + (\mathbf{E}(Y)) + \mathbf{E}(Z))$ and commutativity: $\mathbf{E}(X) + \mathbf{E}(Y) = \mathbf{E}(Y) + \mathbf{E}(X)$. $\square$

### A.4 On the equivalence between formulations of interaction effects

Plugging in the indicator variables into Eq. 4 yields the following equation (writing in matrix form):

$$
\begin{bmatrix} \mathbf{S}(I) \\ \mathbf{S}([X, I]) \\ \mathbf{S}([Y, I]) \\ \mathbf{S}([X, Y, I]) \end{bmatrix} = \begin{bmatrix} 1 & 0 & 0 & 0 \\ 1 & 1 & 0 & 0 \\ 1 & 0 & 1 & 0 \\ 1 & 1 & 1 & 1 \end{bmatrix} \begin{bmatrix} \beta_0 \\ \beta_1 \\ \beta_2 \\ \beta_3 \end{bmatrix} + \epsilon.
$$
(5)

The standard variable elimination operations give us an expression for the expression for the interaction effect parameter $\beta_3$:

$$
\beta_3 = \mathbf{S}([X, Y, I]) - \mathbf{S}([X, I]) - \mathbf{S}([Y, I]) + \mathbf{S}(I),
$$
(6)

which exactly recovers the definition for $\mathrm{Int}(X, Y)$ using the equivalent formulation (Eq. 3).

### A.5 Additional experiment results

Tables 9 – 7 present some additional experiment results.

| Dataset | Reference | Model | Length | Depth | TopConst | BigramShift | Tense | SubjNumber | ObjNumber | OddManOut | CoordInv |
|---|---|---|---|---|---|---|---|---|---|---|---|
| SST2 | *I* | BERT | −4.58*** | 0.3 | −4.94* | −1.86*** | −0.88* | −0.86* | 0.0 | 1.22*** | 0.0 |
| SST2 | COLA | BERT | −4.42* | −1.7 | −2.0 | −0.78* | −0.44 | −0.62 | −0.36 | −0.12 | 1.52 |
| SST2 | MRPC | BERT | −3.92* | −0.02 | −2.2 | −2.42* | 0.2 | −0.26 | −1.62 | −0.88 | 0.16 |
| SST2 | STSB | BERT | −5.28* | −0.6 | −3.44 | −2.68*** | 0.24 | −2.42* | −3.66*** | 0.7 | 2.8* |
| SST2 | QNLI | BERT | −4.22* | −0.28 | −4.88*** | −1.4 | −0.2 | −3.16* | −2.8* | −1.82 | −1.28 |
| SST2 | RTE | BERT | −2.76* | −1.96 | −6.96*** | −2.48*** | −0.24 | −3.82* | −2.0* | −1.38 | −1.3 |
| SST2 | MRPC QNLI | BERT | −2.52 | 0.34 | −4.92* | −2.06* | −0.66 | −1.9 | −1.46 | −1.5 | −0.58 |
| SST2 | MRPC RTE | BERT | −4.52* | −0.84 | −2.5 | −2.18* | −0.58 | 0.5 | −1.58 | −1.38* | −0.36 |
| SST2 | STSB QNLI | BERT | −3.06* | 0.74 | −2.66 | −1.78* | 0.36 | −2.14 | −1.36 | −0.4 | 0.8 |
| SST2 | STSB RTE | BERT | −5.48* | −1.04 | −4.66* | −2.24* | −0.22 | −2.86* | −2.48* | −2.36 | −1.12 |
| SST2 | *I* | RoBERTa | −5.44*** | −2.84* | −5.7*** | −8.76*** | −4.0*** | −7.28*** | −8.06*** | −2.32* | −3.76*** |
| SST2 | COLA | RoBERTa | −8.28*** | −1.9* | −8.14*** | −1.46* | −1.84* | −4.44* | −3.86* | −2.56*** | −1.94 |
| SST2 | MRPC | RoBERTa | −9.18*** | −1.88 | −8.42*** | −3.12 | −2.02 | −2.88 | −4.32*** | −1.44 | −4.26*** |
| SST2 | STSB | RoBERTa | −8.3*** | −2.78* | −5.34* | −2.12 | 0.4 | −3.08 | −4.46*** | −0.88 | −2.32* |
| SST2 | QNLI | RoBERTa | −5.26* | −1.82 | −5.52 | −6.64*** | −1.36 | −4.2 | −4.22* | −1.72 | −0.94 |
| SST2 | RTE | RoBERTa | −5.64*** | −1.36 | 2.44 | −2.98*** | 0.68 | −1.72 | −2.58 | −1.82 | 2.2* |
| SST2 | MRPC QNLI | RoBERTa | −7.1* | −2.26* | −7.94*** | −4.6* | −2.42* | −1.8 | −2.38 | −3.2* | −2.28 |
| SST2 | MRPC RTE | RoBERTa | −4.44* | −0.58 | −6.96* | −4.6* | −1.22 | −6.96* | −4.46* | −1.46 | −3.9 |
| SST2 | STSB QNLI | RoBERTa | −7.84* | −0.86 | −3.5 | −5.92* | −1.7 | −2.36 | −2.42 | −1.96 | −1.94 |
| SST2 | STSB RTE | RoBERTa | −9.76*** | −2.34*** | −9.72* | −4.14* | −2.18 | −4.5 | −2.88 | −1.18 | −1.64 |

Table 9: Individual effects of SST2 dataset with different reference states. *, ** and *** stand for $p < 0.05$, $p < 0.01$, and $p < 0.001$, respectively, for two-sample $t$-test with $dof = 8$.