# OpenReview forum: "A State-Vector Framework for Dataset Effects"
_EMNLP/2023/Conference — EMNLP 2023 Main_

### Official Review · Reviewer_6EYi · 2023-07-22

**Soundness:** 4

**Excitement:**

4: Strong: This paper deepens the understanding of some phenomenon or lowers the barriers to an existing research direction.

**Paper Topic And Main Contributions:**

The paper presents a framework for quantifying the effects of single datasets as well as the interaction effects between sets of datasets for a given model. The framework is formulated as a vector-space, where the i'th dimension of a dataset corresponds to its (general) score on a given probing task.

The strength of this framework lies in its generality, as it does not condense into a singular real-value score for each dataset. Although, I might say that the usage of vector-space is quite redundant in this case as all effects were introduced and computed by very basic linear algebra/probability operations.

The authors conducted experiments on 2 models (bert and roberta), 3 types of datasets, containing 2 datasets per type on 9 probing tasks.
The experiments measured the effect of each dataset as well as the interactions between each pair of datasets on the 9 probing tasks.
The authors included an extensive discussion of the results, highlighting some interesting findings, such as what they refer to as the "spill-over" effect. This phenomenon indicates that certain syntactic datasets exhibit impacts along the semantic dimensions and vice-versa.

**Reasons To Accept:**

The authors introduced a novel vector-based framework aimed at assessing the effects of datasets and their interactions on each other.
The notable advantage of this framework lies in its high level of generality, making it capable of potentially assessing the impacts of a vast array of datasets on different probing dimensions.

The experiments are quite extensive spanning over 2 models, 3 types of datasets, containing 2 datasets per type, and 9 probing tasks.
The authors systematically discuss their findings. Some of them are intriguing, such as the "spill-over" effect that indicates that certain syntactic datasets exhibit impacts along the semantic dimensions and vice-versa. They also discovered that Interaction effects are concentrated in a small number of dimensions and that similar datasets interact less with each other.

Overall, this is a well-constructed paper that has the potential to significantly influence future research in measuring dataset transferability and the impact of a set of datasets on a new collection.

**Reasons To Reject:**

I find the general framework of vector-space to be quite redundant with respect to the conducted experiments. Ultimately, the authors showcase two categories of tables: the first measures the difference in the accuracy of each probing task before and after fine-tuning a model on a specific dataset, while the second gauges the difference between the accuracy of each probing task after jointly fine-tuning a model on two datasets and the sum of accuracies obtained by fine-tuning the model on each dataset separately. The equations that correspond to these two metrics apply basic linear algebra (\probability theory).

Secondly, although the authors used a well-known public probing dataset (SentEval), I found the probing tasks to be rather uninteresting, in the sense that it is quite hard to yield highly impactful findings from them. This is attributed to the limited correlation between the various tasks. As a result, the papers' findings have a broad scope but do not unveil any captivating discoveries within the group of probing tasks.

Within the provided framework, it would have been intriguing to include at least one result on transfer learning, specifically, examining the effect of first fine-tuning a model on one dataset and then fine-tuning it again on a second dataset. I am of the opinion that the paper could gain valuable insights from such a discussion.

**Reproducibility:**

3: Could reproduce the results with some difficulty. The settings of parameters are underspecified or subjectively determined; the training/evaluation data are not widely available.

**Reviewer Confidence:**

4: Quite sure. I tried to check the important points carefully. It's unlikely, though conceivable, that I missed something that should affect my ratings.

**Typos Grammar Style And Presentation Improvements:**

Lines 389-392: "Tables 2 and 3 (in Appendix)... Tables 4 and 8 break down ..." -> only Table 8 appears in the Appendix.

---

> ### Author Rebuttal · Authors · 2023-08-27
>
> Thank you for your reviews! The transfer learning setting is indeed a very interesting scenario. This scenario will be considered in future works. We will fix the typo in the next version.

---

### Official Review · Reviewer_9zjj · 2023-08-01

**Soundness:** 3

**Excitement:**

3: Ambivalent: It has merits (e.g., it reports state-of-the-art results, the idea is nice), but there are key weaknesses (e.g., it describes incremental work), and it can significantly benefit from another round of revision. However, I won't object to accepting it if my co-reviewers champion it.

**Paper Topic And Main Contributions:**

This paper proposes a framework for analyzing the dataset effects, which enables us to investigate if training models on certain dataset(s) influences their linguistic abilities, e.g., sentence length prediction, syntactic tree depth prediction, etc.

The contribution of this paper  is as follows:

- The authors propose a novel framework called State-Vector Framework. The framework allows us to investigate if certain datasets have under-specified (or undesirable) effects on the model's linguistic abilities
- Given the proposed framework, the authors present some interesting findings, e.g., some datasets have negative influence to the abilities that seems unrelated to the task nature.

**Questions For The Authors:**

- Figure 3: What does the "<" shaped line represent?
- plenty of related works for Table 2 and 3?
- L481: "aforementioned confounding variables" to which variables do you refer to? Would you please elaborate?
- L42: "Model developers use large corpora that may include multiple existing datasets (Sun et al., 2021; Wei et al., 2021; Brown et al., 2020)" I believe that the authors are discussing the pretraining phase, in which we mix the corpus from different domains (e.g., Wikipedia, News, and Webtext). Is this correct? If so, this sentence seems to be misleading, because this paper is mostly about finetuning, instead of pretraining.

**Reasons To Accept:**

- I agree that investigating the dataset effect is an important direction. Given that the recent LLMs have billions of parameters, we need a method for investigating the effectiveness of finetuning in advance for computational efficiency.
- The authors have conducted thorough experiments on two language models (BERT and RoBERTa) with some heuristics to reduce the search space.

**Reasons To Reject:**

### Downstream performance is completely ignored

This paper almost completely ignores the downstream performance (e.g., accuracy and F1 score) of finetuning tasks through the entire experiments.
I acknowledge that linguistic abilities are important.
However, I wish to see the discussion on downstream performance and how it relates to the findings in this paper.
For example, does a negative influence on linguistic abilities mean that model performs poorly in the target task?
Is it consistent to the findings of the previous studies such as (Zhu et al, 2022a)?
This aspect is important because the authors regard model developers as one of their targets.
I believe that it is likely that such developers are interested in downstream performance in addition to linguistic abilities.
Please note that I am not asking for the extra experiments; the performance can be reported because the authors have already finetuned the models.

### Variance of Individual Effects and Interactions?

It seems that the authors have conducted the experiments using multiple random seeds. Is this true?
Does it mean that the authors report the average score on Table 6?
If so, I would like to see the variance value in the table.
This is because Figure 1 and 3 seems to present a high variance value; I am highly concerned if the findings in this paper can be reproduced using the other random seeds.

**Reproducibility:**

3: Could reproduce the results with some difficulty. The settings of parameters are underspecified or subjectively determined; the training/evaluation data are not widely available.

**Reviewer Confidence:**

2: Willing to defend my evaluation, but it is fairly likely that I missed some details, didn't understand some central points, or can't be sure about the novelty of the work.

**Typos Grammar Style And Presentation Improvements:**

- If I understand the paper correctly, the act values in Tables 2,3,4,6  are not important. What is important here is the statistical significance. If this is the case, the presentation could be much improved if the authors use heatmap-like representations, in which each cell represents if statistical significance exists or not.
- Please include Table 9 in the main text, instead of burying it into the Appendix. I believe this is necessary to ensure that the paper is self-contained.
- L740: ai --> AI
- Throughout the table: roberta should be RoBERTa and bert should be BERT for consistency.

---

> ### Author Rebuttal · Authors · 2023-08-27
>
> Thank you for your detailed reviews! To the points mentioned in the review:
>
> - You are right in pointing out that downstream performance is not included. The downstream performance itself is not our focus of this paper. In most fine-tuning settings, we combine multiple datasets, so comparing the fine-tuning results to Zhu et al (2022a) would be apple-to-orange.
> - Variance of the results: there are a lot of them; we will include the raw results in the open-sourced github repo. While they appear large on the figures, these variances are considered when conducting the statistical tests (t-tests for individual effects, and ANOVA for interaction effects). When the statistical tests give p<0.05 (those results marked with stars in the tables — there are a lot of them), we consider the variances to be not very large.
> - Figure 3: the “<” shaped segments contain two parts: E(X) and E(Y). The intersection point is S(I).
> - L481: the confounding variables refer to those mentioned in L462. The model choice, hyperparameter selection, etc., can contribute to the confounding.
> - L42: You are right in pointing out that fine-tuning and pre-training are distinct phases. In either phase, people use the combination of multiple datasets. This paper’s experiments only cover the effects in the fine-tuning phase, but the theoretical framework can generalize to the pre-training phase (the difference is that LM pretraining datasets are usually larger, and have more labels than the fine-tuning datasets). We will add this point to the Limitations section.
> - Grammar styles, references to GLUE etc. and presentation points: Thanks. We’ll fix them in the next version.

---

### Official Review · Reviewer_PRSZ · 2023-08-05

**Typos Grammar Style And Presentation Improvements:** Line 585
**Soundness:** 5

**Excitement:**

4: Strong: This paper deepens the understanding of some phenomenon or lowers the barriers to an existing research direction.

**Paper Topic And Main Contributions:**

This paper studies the effect of using multiple datasets on models using a state-vector based probing framework. Such a framework helps in quantifying not only the effect of single or standalone dataset but also interacting datasets. The paper addresses important questions about dataset effects as well as provides how the framework can be useful in curating datasets for responsible model development. Additionally the paper discusses spill-over effects that might creep in via the introduction of the new dataset for a task.

The paper provides comprehensive framework as well as analysis on standard classification tasks using the aforementioned framework.

**Questions For The Authors:**

Line 96-97: What does dataset with complex decision boundaries mean? Ideally a model has decision boundaries.

Table 2,3,4: Are the numbers shown here difference in accuracy levels for each of the probes across 3 epochs?

**Reasons To Accept:**

1. Well written paper with simple mathematically grounded formulation.

2. New Framework for analysing effects of datasets. Such a probing state-vector framework is helpful in understanding the effects of a dataset along multiple dimensions (probes) rather than just having a single “qualitative” score.

3. Comprehensive evaluation on 300 fine-tuning experiments per model.

4. Provides guidelines as to how the introduction of new dataset into the task might affect the performance of the models on different tasks.

**Reasons To Reject:**

1. It would help to show the working of how individual effects were calculated in the tables 2,3,4.

**Reproducibility:**

4: Could mostly reproduce the results, but there may be some variation because of sample variance or minor variations in their interpretation of the protocol or method.

**Reviewer Confidence:**

4: Quite sure. I tried to check the important points carefully. It's unlikely, though conceivable, that I missed something that should affect my ratings.

---

> ### Author Rebuttal · Authors · 2023-08-27
>
> Thank you for the reviews! To the points mentioned in weaknesses and questions:
>
> - Decision boundary: it should be "the optimal models trained on this dataset". We will make the usage of terminology more precise.
> - Tables 2-4 show individual effects, which are the differences in the mean accuracy levels across random seeds. Since there is some randomness in the probing procedure, we take the average of the accuracy to approximate the accuracy of the global optimum.

---

### Meta-Review · Area_Chair_1VGs · 2023-09-19

**Recommendation:** 4

**Metareview:**

This work proposes a framework for analyzing the effect of a dataset on a model's linguistic abilities, as quantified by probing task performance. The framework allows us to investigate if certain datasets have undesirable effects on the model's linguistic abilities. An interesting finding from the work is that some datasets negatively influence abilities that seem unrelated to the task nature.

Overall, the reviewers agree that the soundness of the proposed work is sufficient to support its main arguments, though several reviewers point out additional points that should be addressed in a revision. Specifically, a common note among reviewers is to clarify the results in Tables 2-4 relating to individual effects and display a measure of the variance in a more prominent way. Another limitation that is brought up is that downstream task performance is not reported. Even though downstream task performance is not the focus of the work, it's important to report as the degree of encoded linguistic abilities may relate to the degree of fine-tuning success. We recommend that these points are addressed in a revision. On the positive side, reviewers praised the clarity of the writing and thought the research question to be of significance to the EMNLP audience.

---

### Decision · Program_Chairs · 2023-10-07

**Decision:**

Accept-Main

**Comment:**

This work proposes a framework for analyzing the effect of a dataset on a model's linguistic abilities, as quantified by probing task performance. The framework allows us to investigate if certain datasets have undesirable effects on the model's linguistic abilities. An interesting finding from the work is that some datasets negatively influence abilities that seem unrelated to the task nature.

Overall, the reviewers agree that the soundness of the proposed work is sufficient to support its main arguments, though several reviewers point out additional points that should be addressed in a revision. Specifically, a common note among reviewers is to clarify the results in Tables 2-4 relating to individual effects and display a measure of the variance in a more prominent way. Another limitation that is brought up is that downstream task performance is not reported. Even though downstream task performance is not the focus of the work, it's important to report as the degree of encoded linguistic abilities may relate to the degree of fine-tuning success. We recommend that these points are addressed in a revision. On the positive side, reviewers praised the clarity of the writing and thought the research question to be of significance to the EMNLP audience.